# Fisetin Mitigates Chronic Lung Injury Induced by Benzo(a)Pyrene by Regulation of Inflammation and Oxidative Stress

**DOI:** 10.3390/cimb47030209

**Published:** 2025-03-19

**Authors:** Wanian M. Alwanian

**Affiliations:** Department of Medical Laboratories, College of Applied Medical Sciences, Qassim University, Buraydah 51452, Saudi Arabia; w.alwanian@qu.edu.sa; Tel.: +966-50-314-4654

**Keywords:** apoptosis, Benzo(a)Pyrene, lung cancer, inflammation, oxidative stress, cholesterol, triglycerides, fibrosis

## Abstract

**Background**: Polycyclic aromatic hydrocarbons such as Benzo(a)Pyrene, which are produced by smoking or present in air pollution, greatly contribute to lung diseases. B(a)P has been found to induce inflammation and eventually lung cancer. Fisetin, a polyphenol, abundant in many fruits and vegetables, has an appealing therapeutic potential in many disorders, including inflammation and cancer. **Objectives**: This study aimed to investigate the importance of fisetin in the regulation of chronic lung inflammation and oxidative stress resulting from exposure to Benzo(a)Pyrene. **Methods**: The effect of fisetin on rats at a concentration of 50 mg/kg was evaluated by ELISA to measure oxidative stress and inflammatory markers. The tissue architecture was also investigated using hematoxylin and eosin (H&E) staining. The expression pattern of IL-6 in lung tissues was assayed using immunohistochemistry. Fibrosis was evaluated in lung tissues using Masson Trischrome and Sirius red stains. Cell apoptosis in lung tissues was studied using a TUNEL assay. **Results**: After exposure to Benzo(a)Pyrene for eight weeks, the data indicated that fisetin led to a significant reduction in oxidative stress, evidenced by the reduction of SOD, MDA, NO, GPH, and GPx. Moreover, IL-6, TNF-α, and CRP levels were also decreased, indicating a reduction in inflammation. Apoptosis was reduced upon fisetin treatment. Furthermore, a significant decrease in fibrosis was also observed. **Conclusions**: This study reveals the importance of fisetin as a natural product in the management of chronic lung injury by protecting lung tissues from inflammation, and its use suggests better prognosis in diseases caused by exposure to B(a)P.

## 1. Introduction

Chronic respiratory diseases (CRDs) affect the airways and other structures of the lungs and can progress to lung cancer independent of age, sex, or smoking [1]. CRDs may be caused by tobacco smoking and air pollution [2]. Common types of CRD include asthma, chronic obstructive pulmonary diseases (COPDs), occupational lung diseases, pulmonary hypertension, pneumonia, and lung cancer. A study published in 2023 by The Institute for Health Metrics and Evaluation (IHME) discovered that CRD was globally the third leading cause of mortality, with 4 million deaths and a prevalence of 454.6 million [3]. In 2019, it was also estimated that 434,560 had chronic obstructive pulmonary disease (COPD), accounting for 1.65% of total deaths in Saudi Arabia [4]. Asthma is another prevalent condition in Saudi Arabia and the world, and a growing body of evidence suggests that many cases are underdiagnosed. For instance, a study conducted using 2405 participants, both males and females, between April and June of 2016 in Riyadh argued that the prevalence of asthma was high, with 18% of participants having asthma-related symptoms without reporting cold or flu symptoms. Another survey conducted by Alomary and colleagues using a nationwide cross-sectional study and involving 8955 adults from all regions of Saudi Arabia pointed out that 14.2% suffered from asthma symptoms, among which 38.1% were affected by severe asthma symptoms [5].

Tobacco smoking is one of the risk factors of chronic lung diseases and contributes to the initiation of human cancers [6]. Tobacco smoking kills over 7 million people a year because of first-hand smoking, in addition to 1.7 million deaths because of exposure to second-hand smoking [7]. Tobacco smoking is on the rise between young men and women [8]. In Saudi Arabia, a study performed in 2013 revealed that the prevalence of smoking was 12.2% [9]. Although tobacco smoking is a major contributor to chronic lung diseases, 19% of lung cancer deaths are estimated to be caused by other factors such as air pollution [10].

Fine particulate matter (PM2.5, below 2.5 µm in diameter), which is produced from combustion engines, industrial activities, coal, or cigarette smoke, is classified as a group I human carcinogen [10]. It contains polycyclic aromatic hydrocarbons (PAHs), such as benzo(a)Pyrene (B(a)P), which contribute to the carcinogenic properties of these chemicals [11,12]. B(a)P is generated through the incomplete combustion of organic materials; and when inhaled, it becomes metabolized by cytochrome P450 to produce 7,8-diol-9,10-epoxide-benzo(a)pyrene, a carcinogenic substance [13]. B(a)P causes inflammation, oxidative stress, DNA alteration, cell proliferation, COPD, asthma, and carcinogenesis, eventually leading to millions of annual premature deaths [14]. B(a)P impacts the ability of lung cells to differentiate by influencing lung stem cells [15]. Human lung cells subjected to biomass smoke particles containing B(a)P induce inflammatory responses by releasing IL-6, TNF-α, and IL-8. This activates signaling pathways such as mitogen-activated protein kinase (MAPK) and Toll-like receptors (TLRs) [16,17,18].

Flavonoids are plant-derived and hydroxylated polyphenols isolated from many fruits and vegetables. They have a 15-carbon structure that consists of two phenyl rings joined by a heterocyclic 4H-pyrene ring. Oxidation and substituents added to the 4H-pyrene ring create the diversity of flavonoids [19]. Evidence suggests that flavonoids exert many therapeutic activities such as antioxidative, anti-inflammatory, anti-cancer, anti-proliferative, anti-microbial, and anti-viral [20]. These properties are attributed, in part, to their ability to inhibit NF-κB-related pro-inflammatory cytokines [21].

Fisetin (3,7,3′,5′-tetrahydroxy flavone), present in fruits and vegetables, is one of many flavonoids shown to have potent therapeutic properties. It can easily enter cells through the cell membrane because of its hydrophobic nature [22,23]. Previous studies conducted on rats showed that fisetin downregulated inflammatory responses induced by lipopolysaccharides (LPSs) through the inhibition of NF-κB activation, TLR4 expression, and neutrophil–macrophage infiltration [24]. By NF-κB inhibition, fisetin reduced myocardial injury markers, creatine kinase (CK-MB), and protected the heart’s histology [25]. Furthermore, fisetin treatment inhibited TGF-ß/Smad3 signaling, thereby reducing bleomycin-induced fibrosis (BLM) in mouse lungs [26]. It also decreased the expression of c-myc and cyclin-D1 and inhibited the tumorigenic properties of A549 human cancer cell lines [27]. The proliferation of breast cancer cell lines that over-express EGFR2 or estrogen receptors is also impacted by fisetin treatment [28]. In a study published by Hussain and colleagues on male Wistar rats, fisetin abrogated oxidative stress and inflammation caused by cigarette smoke [29].

Many studies have investigated the importance of fisetin on inflammatory responses and cancer, but the importance of fisetin on chronic lung diseases is still unclear. In this study, we aimed to investigate the therapeutic implications of fisetin against benzo(a)pyrene-induced chronic lung injury.

## 2. Materials and Methods

### 2.1. Antibodies and Reagents

Benzo(a)pyrene was purchased from Sigma-Aldrich (St. Louis, MO, USA). Kits used to estimate antioxidant enzymes and inflammatory markers were purchased from the following vendors: Abcam (Cambridge, UK): catalase (Cat), superoxide dismutase (SOD), Glotathione peroxidase (GPx), interleukin-6 (IL-6), tumor necrosis factor (TNF-α). The following kits were also purchased from Abcam (Cambridge, UK): Masson trichrome and Sirius red, TUNEL assay; lipid peroxidation kits: Malondialdehyde (MDA) and nitric oxide (NO) were purchased from Abcam. Kits used to measure cholesterol and triglycerides were obtained from Crescent Diagnostics.

### 2.2. Development of Chronic Lung Injury Model and Fisetin Treatment

The male Wistar rats (weighing 170–220 g) used in this study were purchased from King Saud University. They had ad libitum access to food and water in a 12 h dark/light cycle and a constant temperature of 23 ± 2 °C. The animals were grouped into A: control (8 rats), B: fisetin treatment only (8 rats), C: disease control (B(a)P treatment) (8 rats), and D: B(a)P + fisetin treatment (8 rats), making a total of 32 rats. Chronic lung injury was induced by oral administration of B(a)P (50 mg/kg) in DMSO twice/week for 8 weeks. For the fisetin-treated group, the rats were treated with 50 mg/kg of fisetin 30 min after b(a)P administration. Ethical procedures were followed in accordance with the “Regulations of Research Bioethics on the Living Creatures”, and ethical approval was obtained by the institution. The animals were fed normal chow and fisetin-based formulations according to their treatment group.

### 2.3. Animal Sacrifice, Lung Isolation, and Blood Collection

All animals were anaesthetized after 8 weeks using chloroform inhalation and then sacrificed. Blood was collected in plain tubes and allowed to clot for 30 min at room temperature; then, it was centrifuged at 3000 rpm for 10 min to collect serum. Serum was aliquoted and stored at −20 °C for further analysis. Lungs were collected at the time of sacrifice, washed in normal saline, and stored in 10% formalin for immunohistochemical analysis or homogenized in potassium phosphate buffer (pH 7.4) for biochemical analysis.

### 2.4. Measurement of Oxidative Stress and Inflammatory Marker Levels

The pro-inflammatory/cytokine and oxidative stress markers were measured in lung homogenates using ELISA kits (Abcam, Cambridge, UK) per the manufacturer’s instructions. The lungs were homogenized in potassium phosphate buffer (pH 7.4). The resulting homogenate was centrifuged at 4 °C for 10 min at 10,000 rpm, and the supernatant was collected and stored at −20 °C. The samples were processed to measure the inflammatory and antioxidant enzymes: SOD, catalase, GPx, GPH, IL-6, TNF-α, and NO activities.

### 2.5. Histopathology Analysis

Lung tissues from all animals were excised at the time of sacrifice for hematoxylin and eosin analysis (H&E) and immunohistochemistry staining. The lung tissue from each animal was immediately placed in 10% formalin, embedded in paraffin, and cut into 5–6 µm sections. Hematoxylin–eosin (H&E) staining was performed to analyze the changes in the tissue’s architecture. Special staining like Masson trichrome stain (Abcam, Cambridge, UK) and Sirius red staining (Abcam, Cambridge, UK) for the assessment of fibrosis was performed and examined under the microscope, and images were taken as previously described [30].

### 2.6. Immunohistochemistry

IL-6 protein expression was detected using immunohistochemistry (IHC) by the earlier described method [31,32] using a specific antibody. Paraffin-embedded tissues were sectioned, and deparaffinization was performed using xylene treatment. The slides were immersed in a retrieval solution and heated in a water bath to retrieve the masked antigens. Then, sections were exposed by hydrogen peroxide to block endogenous peroxidase, followed by rinsing in a phosphate-buffered solution and pre-incubation in bovine serum albumin (BSA) to block non-specific binding sites. The slides were incubated with the primary antibody for 45 min. Then, they were incubated with a secondary antibody for 30 min and tertiary antibodies for 15 min. The expression of IL-6 protein was visualized using diaminobenzidine (Abcam, Cambridge, UK). The slides were then counterstained with hematoxylin.

### 2.7. Evaluate Apoptosis Through TUNEL Assay (Terminal Deoxynucleotidyl Transferase-Mediated dUTP Nick-End Labeling Assay)

To evaluate apoptosis in tissues, a TUNEL assay was performed following the kit’s manufacturer’s instructions. The apoptotic index was calculated by determining the percentage of TUNEL-positive cells. For each tissue, 5 high power fields were evaluated [33].

### 2.8. Statistical Analysis

Analyses were conducted using SPSS for Windows (v. 15.0). Data were compared with diseased controls or negative controls using multivariate analysis of variance. The *p* values less than 0.05 were considered as statistically significant.

## 3. Results

### 3.1. Fisetin Attenuates Oxidative Stress Following Benzo(a)Pyrene Treatment

To develop the animal model, rats were treated with either vehicle (control), B(a)P, fisetin + B(a)P together, or fisetin alone for 8 weeks before they were sacrificed. To assess the ability of fisetin to counteract the oxidative stress effects of benzo(a)Pyrene, we investigated the levels of antioxidative enzymes and oxidative markers. Our data show that rats exposed to B(a)P had upregulation in Malondialdehyde (MDA) and nitric oxide (NO) compared with the control animal group, indicating an increase in oxidative stress and lipid peroxidation. In comparison, the fisetin-treated group, after B(a)P exposure, showed significant downregulation of MDA and NO (Figure 1A,B). Next, we sought to determine the levels of antioxidative enzymes: superoxide dismutases (SODs), catalase (CAT), Glutathione Peroxidase (GPx), and Glutathione (GSH), and we found that upon treatment with B(a)P, these enzymes were upregulated, suggesting oxidative stress. However, the upregulation of these enzymes, which was caused by B(a)P, was significantly diminished by fisetin treatment compared with the control or fisetin treatments (Figure 1C–F). These data suggest that fisetin may reduce oxidative stress caused by benzo(a)Pyrene.

### 3.2. Fisetin Treatment Downregulates Inflammation and the Increase in Lipids Induced by B(a)P

B(a)P treatment induces inflammation in mouse lungs [34]. To determine if fisetin has a positive impact on inflammation, we assayed for the inflammatory markers, IL-6, TNF-α, and C-reactive protein (CRP). Both IL-6 and TNF-α have broad functions as proinflammatory cytokines, and they are initiated first in inflammation [35,36]. CRP is also a mediator in inflammation, and its synthesis is facilitated by IL-6 [37,38]. By performing ELISA assays on lung tissue homogenates collected from animal groups, we showed that, indeed, animals that were treated with B(a)P had a significant increase in inflammatory response, evidenced by the increase in IL-6, TNF-α, and CRP, and this increase was significantly lessened by fisetin treatment (Figure 2A–C). We also investigated IL-6 expression in lung tissue by immunohistochemistry, and we found that IL-6 expression is consistent as it is increased in the lungs of rats that were exposed to B(a)P alone, and this expression was reduced by fisetin treatment (Figure 2D).

Multiple reports have indicated that there is a correlation between inflammation and plasma lipids, and that the inflammation can be a risk factor for the increase in plasma lipid levels [39,40]. Moreover, low levels of CRP can determine if patients would benefit from a low fat diet [41]. So, we looked at the plasma levels of cholesterol and triglycerides in rats treated with B(a)P alone, B(a)P + fisetin together, or fisetin alone, and when compared with the control, we found that B(a)P caused an increase in the levels of cholesterol and triglycerides. However, treatment with fisetin led to a significant decrease in plasma lipid levels (Figure 2E,F). This suggests that fisetin can prevent inflammation, and consequently, the rise in plasma lipid levels that can be induced by B(a)P.

### 3.3. Fisetin Alleviates Lung Tissue Damage Induced by Benzo(a)Pyrene

B(a)P is a well-studied compound that can influence lungs and causes tissue damage [14]. To determine if fisetin can prevent damage to lungs that may be caused by B(a)P, we investigated the lung tissue architecture using H&E staining. We did not observe any changes in the architecture of the lung tissue in the control group. In the B(a)P group, congestion, fibrosis, and infiltration of inflammatory cell were observed. However, treatment with fisetin led to the abolishment of tissue damage (Figure 3A). In acute and chronic lung diseases, lung alveolar tissue remodeling occurred, which led to the increase in collagen fiber [42]. So, to assess how severe tissue damage occurred, we performed staining with Masson trichrome and Sirius red stains to delineate collagen fiber deposition in lung tissues. In the control group, we observed normal levels of collagen fibers. However, the B(a)P group carried high levels of collagen fibers, while in the B(a)P + fisetin group, we observed a decrease in collagen fiber deposition (Figure 3B,C), suggesting a protective role for fisetin against lung fibrosis caused by B(a)P.

### 3.4. Fisetin Is Protective Against Apoptosis Caused by Benzo(a)Pyrene

Since we have shown that B(a)P led to an increase in oxidation and antioxidant enzymes, which can eventually lead to apoptosis, we investigated if there were apoptotic cells in the lung tissues of animals treated with B(a)P and if fisetin could be of protective value against cell death. We did not observe any TUNEL-positive cells in the control group. However, the B(a)P group showed TUNEL-positive cells, which is an indication of apoptosis. Interestingly, animals that were given B(a)P and then fed fisetin had fewer apoptotic cells (Figure 4). Altogether, these data indicate that fisetin may protect lung cells from apoptotic death.

## 4. Discussion

The treatment of chronic lung injury is a challenge, considering its high incident rate. This work illustrated the in vivo therapeutic potential of fisetin, a flavonoid, in chronic lung injury caused by B(a)P using male Wistar rats as the model. B(a)P affects respiratory airways and lung tissues, causing inflammation that may eventually progress to tumorigenesis [13]. Patients with chronic lung inflammation do suffer from irreversible changes that progress to the development of pulmonary fibrosis. Additionally, lung inflammation results in alveolar epithelium damage and pulmonary edema [43]. Many flavonoids have been reported in preclinical studies to have therapeutic effects by modulating oxidative stress and inflammation [20]. Quercetin, Rutin, and Apigenin decrease the production of TNF-α, IL-1β, and IL-6 [44,45,46]. Nonetheless, not many studies have explored the effects of flavonoids on chronic lung injury. Here, we demonstrate that fisetin, a flavonoid, has anti-oxidative and anti-inflammatory functions and can protect lung tissue from death and damage.

The literature reports that B(a)P initiates lung injury by inducing oxidative stress [47]. Alveolar cells express antioxidative enzymes that promote redox reactions and neutralize ROS, leading to tissue maintenance. ROS causes severe outcomes in the lungs due to their high oxygen intake, and, in turn, defense mechanisms such as SOD enzymes are upregulated. ROS can also induce lipid peroxidation and produce highly active aldehydes such as MDA that can react with biomolecules, thereby changing their structure and inducing damage [48]. In this study, we found that when rats were treated with B(a)P, the expression of the oxidative markers MDA, NO, SOD, GPx, and GSH was increased, which suggests high oxidation, and if not treated, it can lead to tissue damage. However, the oxidative stress was mitigated by fisetin treatment. Many hydroxyl groups are present in the chemical structure of fisetin, which allows it to scavenge ROS such as superoxide anions (O^2−^) and hydrogen peroxide (H_2_O_2_) [49], providing one explanation of the antioxidative properties of fisetin. This is also observed in other flavonoids since they share a lot of their structural characteristics. However, the number of these hydroxyl groups is not as important as their position in the chemical structure [20]. For instance, published data indicate that isovitexin and bicalin, two flavonols, differ in their hydroxyl content. Isovitexin has three hydroxyl groups while bicalin bears two hydroxyl groups, but bicalin scavenges DPPH radicals while isovitexin does not [50].

During oxidative stress, flavonoids indirectly prohibit oxidative stress by activating the nuclear factor Nrf-2 and mediating its translocation to the nucleus. Then, it binds the cis-acting antioxidant response elements (AREs), directing the gene expression of antioxidative proteins [51]. In one study performed by Li Zhange and colleagues, fisetin was ineffective in preventing oxidative stress following traumatic brain injury in mice lacking Nrf-2 gene (Nrf-2^−/−)^ [52]. Another study reported that fisetin enhanced the nuclear fraction of Nrf-2 [53]. This also provides another explanation of how fisetin might protect against oxidative stress.

In inflammation, IL-6, TNF-α, and CRP are highly expressed and modulate inflammatory response. Crosstalk between TNF-α and other inflammatory markers such as NF-κB and CRP exists, in which TNF-alpha has to bind its receptor in order for NF-κB to be activated [54]. In acute inflammation induced by LPS, fisetin alleviated inflammatory responses and improved lung tissue architecture [24]. Our data were consistent as they indicated that fisetin inhibited the upregulation of TNF-α and reduced inflammation caused by B(a)P. Other flavonoids such as rutin, apigenin, vitexin, and diosmin were able to inhibit TNF-α that was induced by LPS and subsequently reduce inflammation [55,56,57,58].

Interestingly, evidence from different studies suggests that crosstalk between inflammation and lipoproteins may exist [39,40,41]. It has been shown that during inflammation, hepatic production of cholesterol and fatty acids is stimulated by IL-6 [59], suggesting a role for inflammation in the regulation of many systematic organ diseases. Hence, we measured cholesterol and triglycerides levels in blood samples from the animal model tested, and we observed an increase in lipids investigated due to treatment with B(a)P. However, fisetin treatment managed to reduce cholesterol and triglycerides significantly.

The histopathological architecture of the lung is altered upon exposure to B(a)P, leading to fibrotic changes [42,43]. In the control rats, normal morphology and the architecture of the lung were noted. However, an investigation of the lung tissues exposed to B(a)P for 8 weeks indicated injury that led to inflammation, cell infiltration, fibrosis, and hemorrhage. This is consistent with repercussions that occur due to the increase in ROS production and inflammatory response, leading to damage in the lung microenvironment. Typically, lung inflammation precedes the development of fibrotic changes [43]. An analysis of lung tissues stained with special stains such as Masson trichrome and Sirius red showed an increase in collagen fibers, suggesting damage that may impair function. However, fisetin was able to prevent tissue damage and promote survival by blocking apoptosis. Zhang and colleagues provided evidence that fisetin reduces the senescence of alveolar cells and the trans-differentiation of fibroblasts to myofibroblasts by the inhibition of TGF-ß/Smad3 [26]. Quercetin has also been shown to protect lung tissues from collagen deposition either alone or in combination with dasatinib [60,61]. Collectively, these data discussed here provide evidence for the therapeutic value that can be provided by flavonoids such as fisetin.

Due to the severe health outcomes that chronic lung inflammation and fibrosis cause, there is crucial need for effective therapeutic approaches. Existing treatments can either temporarily hinder disease promotion or cause severe side effects. Fisetin has shown promise, as reported by many studies, in multiple types of diseases including cancer. Consequently, the use of fisetin along with other lines of therapy could prove to be beneficial.

## 5. Conclusions

Fisetin is a polyphenol that is abundant in many fruits and vegetables. It has been shown to have many therapeutic benefits, including anti-inflammatory, anti-cancer, and antioxidative characteristics. We demonstrated the ability of fisetin to alleviate chronic lung inflammation and oxidative stress induced by Benzo(a)pyrene. These data show the therapeutic potential of fisetin and suggest that it may be beneficial in combination with other therapies for chronic lung disease. Further studies should be performed to study the effect of fisetin on other markers of inflammation. Also, the definition of the toxicity level, dose, and mechanism of action in the management of lung pathogenesis would qualify fisetin to be explored in clinical trials.

## Figures and Tables

**Figure 1 cimb-47-00209-f001:**
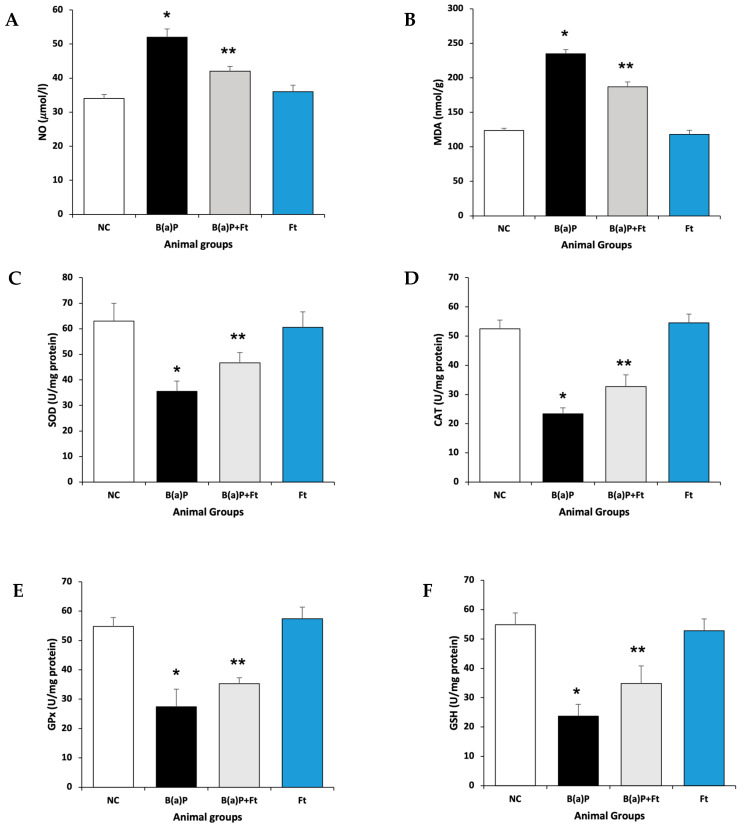
Effect of fisetin on lipid peroxidation and antioxidative enzymes. (**A**) MDA and (**B**) NO levels in different treatment animal groups. The antioxidant enzymes’ potential was measured in different groups: NC (negative control); B(a)P (Benzo(a)Pyrene); B(a)P + fisetin (Ft); or fisetin alone (Ft), as (**C**) SOD, (**D**) catalase (CAT), (**E**) GPx, and (**F**) GSH. The numbers signify the mean ± SEM with 8 animals per group. The statistical differences are represented with an asterisk (*) indicating significance at (*p* < 0.05) as compared with the control group and double asterisks (**) indicating (*p* < 0.05) when compared with the B(a)P-treated group.

**Figure 2 cimb-47-00209-f002:**
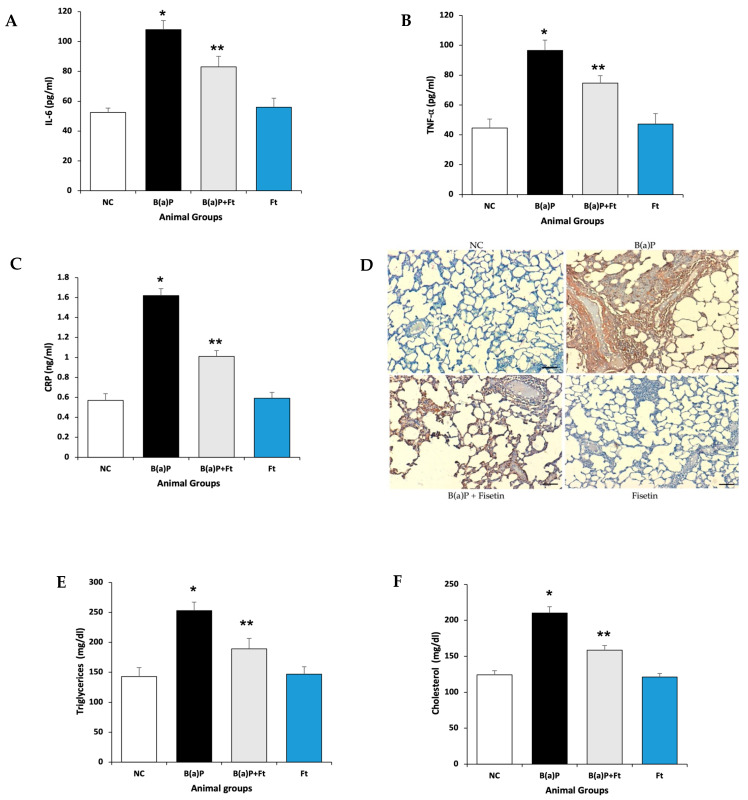
Effect of fisetin on inflammatory markers, cholesterol, and triglycerides. ELISA analysis on inflammatory markers and lipids in different animal groups: NC (negative control); B(a)P (Benzo(a)Pyrene); B(a)P + fisetin (Ft); or fisetin alone (Ft). (**A**) IL-6, (**B**) TNF-α, (**C**) CRP. (**D**) The immunohistochemical analysis of IL-6 protein expression. In the NC group, IL-6 protein expression did not occur. In the B(a)P-treated group, the IL-6 protein expression was high in the cytoplasm. In the B(a)P + fisetin (Ft), the expression of IL-6 was reduced. In the fisetin (Ft)-treated group, no expression of IL-6 was observed. (Scale bar = 50 μm.) (**E**) Triglyceride, (**F**) cholesterol. The numbers signify the mean ± SEM with 8 animals per group. The statistical differences are represented with an asterisk (*) indicating significance at (*p* < 0.05) as compared to the normal control group and double asterisks (**) indicating (*p* < 0.05) as compared with the B(a)P-treated group only.

**Figure 3 cimb-47-00209-f003:**
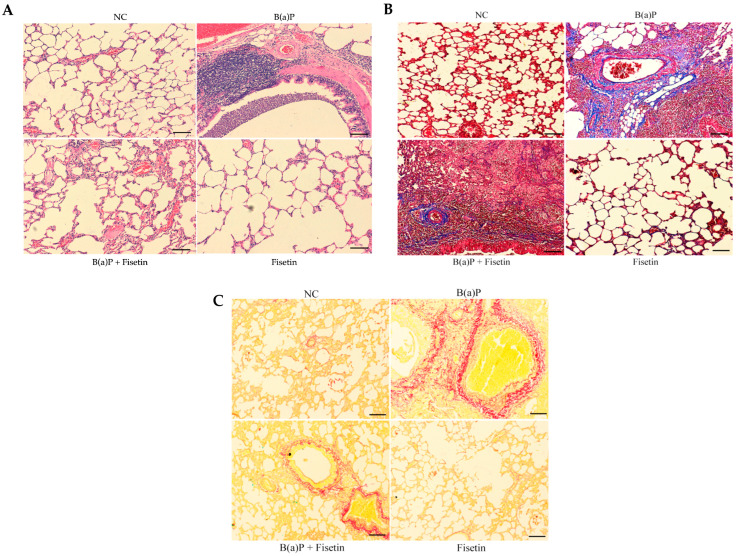
The role of fisetin on lung tissue architecture. Animal lungs were removed, cut into a 5 µm thickness according to different animal groups, and stained with (**A**) H&E staining: in the NC (negative control), the lung tissue architecture was normal. In the B(a)P (Benzo(a)Pyrene) group, congestion and infiltration of inflammatory cells were noticed. In the B(a)P + fisetin (Ft) group, the histopathological changes that were caused by B(a)P were reduced. In the group of fisetin alone (Ft), lung tissue architecture was normal. (**B**) Masson trichrome staining: in the NC, no fibrosis was detected. In the B(a)P group, a thick bundle of collagen fiber was noted (blue color). In the B(a)P+ fisetin group, fibrosis was reduced (blue color). In the fisetin group, no fibrosis was detected. (**C**) Sirius red staining: in the normal control (NC), no fibers accumulation were detected. In the B(a)P group, accumulation of thick bundle of fiber was observed (red color). In the B(a)P+ fisetin group, fibrosis was reduced (red color). In the fisetin group, no fibrosis was detected (scale bar = 50 μm).

**Figure 4 cimb-47-00209-f004:**
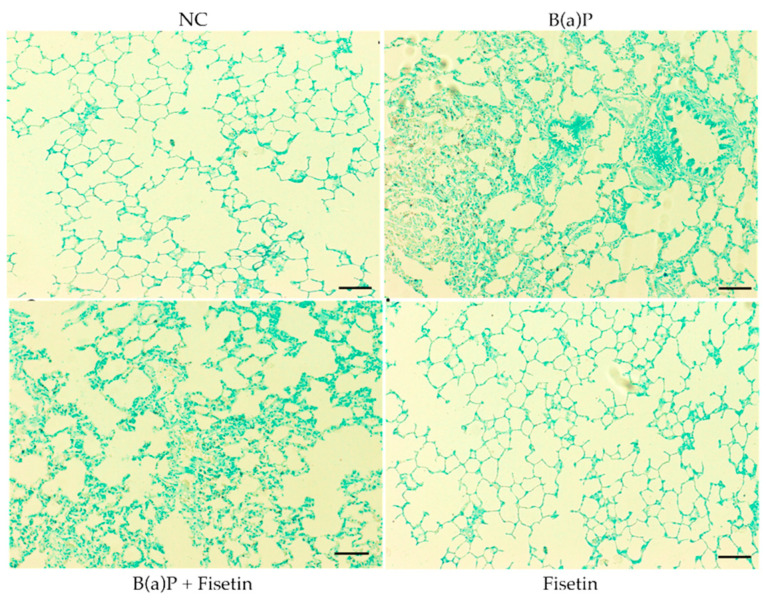
The effect of fisetin on apoptosis using TUNEL staining. DNA fragmentation was assayed and evaluated according to protocol. NC (negative control): no apoptotic body was present. In the B(a)P group, apoptotic bodies were abundant (dark brown). In the B(a)P + fisetin (Ft) group, apoptotic bodies were reduced. In the group of fisetin alone (Ft), no apoptotic body was seen (scale bar = 50 μm).

## Data Availability

The manuscript contains all data described within the text.

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
