# Peer review of "Fisetin Mitigates Chronic Lung Injury Induced by Benzo(a)Pyrene by Regulation of Inflammation and Oxidative Stress"

_cimb, 2025, doi:10.3390/cimb47030209_

Round 1

Reviewer 1 Report

Comments and Suggestions for Authors

Peer review of the article "Fisetin mitigates chronic lung injury induced by benzo(a)pyrene by regulation of inflammation and oxidative stress"

The study addresses a significant public health concern related to lung diseases caused by environmental pollutants. The focus on fisetin as a natural therapeutic agent is timely, given the increasing interest in plant-derived compounds for disease management. The study follows a well-structured experimental design, including control and treatment groups. The use of multiple biochemical assays (ELISA, histopathology, immunohistochemistry, TUNEL assay) strengthens the reliability of the findings.

Ethical considerations are explicitly stated, ensuring compliance with research bioethics.

The inclusion of statistical analysis provides robustness to the study. Figures and tables effectively present key findings. The discussion connects the findings with prior research, demonstrating an understanding of the field.

My comments:

1. Some sentences are lengthy and could be restructured for better readability.

2. The study provides strong evidence that fisetin reduces oxidative stress and inflammation, but the molecular pathways remain insufficiently explained. Additional discussion on the specific signaling pathways influenced by fisetin (e.g., NF-κB, Nrf2) would enhance the mechanistic understanding

3. The paper states that p-values < 0.05 are significant but does not consistently provide exact p-values. Indicating effect sizes or confidence intervals would be beneficial to better interpret the data.

4.   The figures lack clear legends, making it difficult to interpret the data at first glance. The resolution of images could be improved for better clarity.
5.    The conclusion could be expanded to suggest next steps, such as testing fisetin in human clinical trials or evaluating its effects on other markers of lung injury.

Author Response

Comment1:   Some sentences are lengthy and could be restructured for better readability.

Response 1: Thank you for the valuable suggestion. I agree with this suggestion and Sentences has been restructured accordingly throughout the manuscript. You can see examples in pages 2 and 3 (highlighted in red)

Comment2: The study provides strong evidence that fisetin reduces oxidative stress and inflammation, but the molecular pathways remain insufficiently explained. Additional discussion on the specific signaling pathways influenced by fisetin (e.g., NF-κB, Nrf2) would enhance the mechanistic understanding

Response2: Agree. this comment has been addressed in the discussion and more details were added to include NF-κB and Nrf2 and how they might be impacted by fisetin. Please see that in the discussion - Page 9- third paragraph for Nrf-2 and Page 9 – fourth paragraph for NF-κB.

Comment 3: The paper states that p-values < 0.05 are significant but does not consistently provide exact p-values. Indicating effect sizes or confidence intervals would be beneficial to better interpret the data.

Response3: The p values were slightly less that 0.05 and ranged from 0.035 to 0.042, and for clarity, we used P<0.05.

Comment 4: The figures lack clear legends, making it difficult to interpret the data at first glance. The resolution of images could be improved for better clarity.

Response 4: I Agree. this comment has been addressed and figure legends were modified, and more explanation and details of the experiments have been added. Legends for figures 3 in page 7 and figure 4 in pages 8. The image clarity was also improved.

Comment 5: The conclusion could be expanded to suggest next steps, such as testing fisetin in human clinical trials or evaluating its effects on other markers of lung injury.

Response 5: I agree with that and the conclusion has been modified according to the this comment and more future directions were highlighted in red in page 10.

Reviewer 2 Report

Comments and Suggestions for Authors

The abstract is well written and covers all parts of the work.

The introduction addresses the main points of the work, in addition to using current references.

The objectives are clear and well defined, although fisetin is widely studied for fighting cancer, there is only one article that tested the antioxidant and anti‐inflammatory potential of the plant flavonoid, fisetin against cigarette smoke‐induced oxidative stress, and inflammation in rat lungs (Hussain, et al. 2019). However, this work is much more complete.

Tajamul Hussain, Omar S. Al‐Attas, Salman Alamery, Mukhtar Ahmed, Hamza A. M. Odeibat, Salman Alrokayan. The plant flavonoid, fisetin alleviates cigarette smoke‐induced oxidative stress, and inflammation in Wistar rat lungs. J Food Biochem. 2019;43:e12962. https://doi.org/10.1111/jfbc.12962

For the material and methods, I suggest adding the references on which the experiments were based.

The results were well presented, the figures have captions and are important for understanding the paper.

I suggest that the discussion be rewritten, there are several parts that could be removed because they were already presented in the introduction. I believe that seeking comparisons with other flavonoids chemically similar to fisetin could enrich the work.

Overall, the article is original, high quality and well written, therefore, I suggest that the journal accept it with revisions.

Author Response

Comment 1: The objectives are clear and well defined, although fisetin is widely studied for fighting cancer, there is only one article that tested the antioxidant and anti‐inflammatory potential of the plant flavonoid, fisetin against cigarette smoke‐induced oxidative stress, and inflammation in rat lungs (Hussain, et al. 2019). However, this work is much more complete.

Tajamul Hussain, Omar S. Al‐Attas, Salman Alamery, Mukhtar Ahmed, Hamza A. M. Odeibat, Salman Alrokayan. The plant flavonoid, fisetin alleviates cigarette smoke‐induced oxidative stress, and inflammation in Wistar rat lungs. J Food Biochem. 2019;43:e12962. https://doi.org/10.1111/jfbc.12962

Response 1: Thank you for the valuable suggestions. I red the article and it was cited in the article to enrich and work presented. Please see that highlighted at the end of the introduction in page 2.

Comment 2: For the material and methods, I suggest adding the references on which the experiments were based.

Response 2: I agree with this valuable suggestion and this comment has been addressed and more references for materials and methods have been added which should enrich this section. Please see them highlighted in red.  

The results were well presented, the figures have captions and are important for understanding the paper.

Comment 3: I suggest that the discussion be rewritten, there are several parts that could be removed because they were already presented in the introduction. I believe that seeking comparisons with other flavonoids chemically similar to fisetin could enrich the work.

Response 3: I agree. This comment was addressed, and the discussion has been modified and improved. More details and discussions were written. Comparisons of fisetin with other flavonoids has been added and the whole discussion was highlighted because it was completely rewritten. Please see that at page 9.

Reviewer 3 Report

Comments and Suggestions for Authors

I have several concerns regarding the manuscript titled “Fisetin mitigates chronic lung injury induced by Benzo(a)Pyrene by regulation of inflammation and oxidative stress”.

As the objective of the manuscript was to investigate effects of fisetin in the regulation of chronic lung inflammation resulting from exposure to benzo[a]pyrene, I am not sure that the model of oral application of BaP is appropriate. I am aware that this model is used in various studies, but given that BaP is part of cigarette smoke and air pollution, inhalation/intratracheal model would be more desirable.

Please include data regarding animal strain (this is mentioned in Discussion solely), and dose of fisetin (mentioned in abstract) in the Material and method section. Also, there are no data regarding CRP and plasma lipids measurements.

Explain why the 50 mg/kg fisetin dose was chosen, and how the animals were treated.

Why were oxidative stress and inflammation measured in plasma? These parameters can be determined in lung tissue homogenates. Only relevant data regarding lungs are histopathology and immunohistochemistry. Other data describe the systemic effect of BaP.

Although fisetin mitigated the measured effects of BaP, it did not completely restore values to control levels. Would a higher dose of fisetin be more beneficial?

Please discuss obtained results appropriately. There is no need to recapitulate results in Discussion section.

Author Response

Comment 1: As the objective of the manuscript was to investigate effects of fisetin in the regulation of chronic lung inflammation resulting from exposure to benzo[a]pyrene, I am not sure that the model of oral application of BaP is appropriate. I am aware that this model is used in various studies, but given that BaP is part of cigarette smoke and air pollution, inhalation/intratracheal model would be more desirable.

Response 1: Thank you for this valuable question. The method of delivery was based on literature and guarantees controlled delivery and dose of the substance being tested, i.e. B(a)P. Cigarette smoke inhalation contains more substances and carcinogens and does not indicate the exact cause of the effects being observed.  Additionally, exposure to smoke in research setting has faced many challenges due to the uncontrolled doses that are delivered to subjects such as mice and rats.

Comment 2: Please include data regarding animal strain (this is mentioned in Discussion solely), and dose of fisetin (mentioned in abstract) in the Material and method section. Also, there are no data regarding CRP and plasma lipids measurements.

Response 2: I agree. this comment has been addressed and rat strain as well as dose of fisetin were properly explained in materials and methods in page 3 section 2.2.. (highlighted in red).

Comment 3: Explain why the 50 mg/kg fisetin dose was chosen, and how the animals were treated.

Response 3: I appreciate this great comment. this concentration is well-tested in literature and assures minimal toxicity. The animals were treated with fisetin 30 minutes after they were given B(a)P – this has been added to the materials and methods and figure legends in page 3 section 2.2.. (highlighted in red).

Comment 4: Why were oxidative stress and inflammation measured in plasma? These parameters can be determined in lung tissue homogenates. Only relevant data regarding lungs are histopathology and immunohistochemistry. Other data describe the systemic effect of BaP.

Response 4: I truly appreciate this comment from the reviewer since I discovered that these details were mis-explained in the text. The oxidative stress and inflammation were indeed measured from lung tissue homogenates and not plasma. I think that the stress of writing the manuscript had let to this mistake. It has been modified in the text and I appreciate the reviewer’s duo diligence. Changes were made in materials and methods in section 2.3 and in the results (section 3.2).

Comment 5: Although fisetin mitigated the measured effects of BaP, it did not completely restore values to control levels. Would a higher dose of fisetin be more beneficial?

Response 5: Increasing fisetin concentration raises concerns regarding toxicity and since most of published literature used 50 mg/kg, it was used in this study, and I believe using more concentration should be backed by more research.

Comment 6: Please discuss obtained results appropriately. There is no need to recapitulate results in Discussion section.

Response 6: I agree. Discussion has been already modified to include more clarity and ideas and results recapitulation has been removed. Please see that in the discussion section in page 9.

Round 2

Reviewer 1 Report

Comments and Suggestions for Authors

I have no comments.

Reviewer 3 Report

Comments and Suggestions for Authors

I have no additional comments.